# An SNR-Optimized Scanning Strategy for the Geostationary Carbon Cycle Observatory (GeoCarb) Instrument

Jeffrey Nivitanont[1], Sean M. R. Crowell[2], and Berrien Moore III[2]

[1]Department of Mathematics, University of Oklahoma, Norman, OK, USA
[2]College of Atmospheric and Geographic Sciences, University of Oklahoma, Norman, OK, USA

**Correspondence:** Jeffrey Nivitanont (jeffniv@ou.edu)

**Abstract.** The Geostationary Carbon Cycle Observatory (GeoCarb) will make measurements of greenhouse gases over the contiguous North and South American land masses at daily temporal resolution. The extreme flexibility of observing from geostationary orbit induces an optimization problem, as operators must choose what to observe and when. The proposed scanning strategy for the GeoCarb mission tracks the sun's path from East to West and covers the entire area of interest in five coherent regions in the order of Tropical South America East, Tropical South America West, Temperate South America, Tropical North America, and Temperate North America. We express this problem in terms of a geometric set cover problem, and use an incremental optimization (IO) algorithm to create a scanning strategy that minimizes expected error as a function of the signal-to-noise ratio (SNR).

The IO algorithm used in this studied is a modified Greedy algorithm that selects, incrementally at 5-minute intervals, the scanning areas with the highest predicted SNR with respect to airmass factor (AF) and solar zenith angle (SZA) while also considering operational constraints to minimize overlapping scans and observations over water. As a proof of concept, two experiments are performed applying offline the IO algorithm to create an SNR-optimized strategy and compare it to the proposed strategy. The first experiment considers all land masses with equal importance and the second experiment illustrates a temporary campaign mode that gives major urban areas greater importance weighting. Using a simple instrument model, we found that there is a significant performance increase with respect to overall predicted error when comparing the algorithm-selected scanning strategies to the proposed scanning strategy.

## 1 Introduction

Understanding the effects of anthropogenic carbon dioxide ($CO_2$) on the carbon cycle requires us to understand the spatial distribution of atmospheric $CO_2$ concentrations to identify natural and anthropogenic sources and sinks. In addition to a sparse in situ sampling network, ground-based remote sensing measurements are currently obtained from the Total Column Carbon Observing Network (TCCON) and space-based measurements from the Orbiting Carbon Observatory (OCO-2) (Eldering et al. (2017a), Eldering et al. (2017b), Crisp et al. (2017), Crisp et al. (2008), Crisp et al. (2004)) and Greenhouse Gases Observing Satellite (GOSAT) (Kuze et al. (2009), Yokota et al. (2009), Hammerling et al. (2012)). These instruments have provided a wealth of data for understanding the global carbon cycle in recent years. However, these instruments have spatial and temporal

limitations. The repeat cycles of the space-based instruments force the spatial and temporal interpolation of the atmospheric $CO_2$ concentrations within their respective cycles, 3 days for GOSAT (Kuze et al. (2009)) and 16 days for OCO-2 (Miller et al. (2007)). The sparsity of the TCCON measurement sites restricts the latitudinal range of observations. The new Geostationary Carbon Cycle Observatory (GeoCarb) (Moore et al. (2018), Polonsky et al. (2014)) will provide measurements that augment 5 the current remote sensors on the ground and in space in both temporal and spatial coverage.

Recently selected as NASA's Earth Venture Mission-2 (EVM-2), GeoCarb is set to launch into geostationary orbit in 2022 to be positioned at approximately $85°$ ($\pm 15°$) West longitude, with the mission of improving the understanding of the carbon cycle. Building on the work of OCO-2, GeoCarb will observe reflected sunlight daily over the Americas, and retrieve the column average dry air mole fraction of carbon dioxide ($XCO_2$), carbon monoxide (XCO), methane ($XCH_4$), and solar-induced 10 fluorescence (SIF). Moore et al. (2018) identify six major hypotheses about the Carbon-Climate connnection that the GeoCarb mission aims to provide insight on: (1) The ratio of the $CO_2$ fossil source to biotic sink the conterminous United States (CONUS) is ∼4:1, (2) Variation in productivity controls the spatial pattern of terrestrial uptake of $CO_2$, (3) The Amazonian Forest is a significant (0.5–1.0 PgC/year) net terrestrial sink for $CO_2$, (4) Tropical Amazonian ecosystems are a large (50–100 PgC/year) source for $CH_4$, (5) The CONUS methane emissions are a factor 1.6 ± 0.3 larger than in the EPA database, (6) 15 Larger cities are more CO2 emissions efficient than smaller ones. These six hypothesis were used as a basis to select the ∼$85°$W observing slot as the position with most "potential for significant scientific advances."

GeoCarb will view reflected sunlight from Earth through a narrow slit that projects on the Earth's surface to an area measuring about 1,690 miles (2,700 kilometers) from north to south and about 3.2 miles (5.2 kilometers) from east to west. The instrument will make measurements along the slit with a ∼9 second integration time. Instrument pointing will be accom- 20 plished by way of two scanning mirrors that shift the field of view north-south and east-west. The pointing system is extremely flexible, and observations can be made at any location and time with sufficient solar illumination. This flexibility induces an optimization problem: Where should the instrument take measurements at a given time throughout the day?

Determining when and where to make daily scans with GeoCarb's observing capabilities is mathematically similar to a $CO_2$ ground observation network optimization problem for establishing new observation sites. Selecting the optimal location of 25 new observing stations has been shown to be feasible by utilizing various optimization algorithms. There have been previous studies performed on the problem of optimizing $CO_2$ observation networks utilizing computationally expensive evolutionary algorithms [i.e., Simulated Annealing (Rayner et al. (1996); Gloor et al. (2000)) and Genetic Algorithm (Nickless et al. (2018))] and one utilizing a deterministic, incremental optimization (IO) algorithm (Patra and Maksyutov (2002)). All of the previous studies mentioned employed their optimization routines to minimize $CO_2$ measurement uncertainty as a function of signal-to- 30 noise ratio (SNR).

In this paper, a deterministic, IO routine is utilized to find a geostationary scanning strategy that minimizes GeoCarb's expected $CO_2$ measurement uncertainty as a function of SNR for the satellite viewing area. Section 2 gives background information on the GeoCarb mission and the objectives for this paper. Section 3 explains the process used to create the SNR-optimizing, IO algorithm and how the expected error is calculated from the simulated retrievals. In Section 4, a comparison is done between 35 an algorithm-selected strategy and the baseline strategy in the case where all American land masses between $50°$N and $50°$S

are scanned with equal importance weighting. In Section 5, a case study is performed to exhibit a "city campaign" mode for the IO algorithm. We offer concluding statements in Sect. 6 and future research goals.

## 2 Background

GeoCarb will be hosted on a SES Government Solutions (http://www.ses-gs.com) communications satellite in geostationary orbit at $\sim 85°$W. It will measure reflected sunlight in the $O_2$ band at $0.76\mu$m to measure total column $O_2$, the weak and strong $CO_2$ bands at $1.61\mu$m and $2.06\mu$m to measure $XCO_2$, and the $CH_4$ band at $2.32\mu$m for measuring $XCH_4$ and XCO. The $O_2$ spectral band allows for determination of mixing ratios and the measurement of SIF, as well as additional information on aerosol and cloud contamination of retrievals. The baseline mission for GeoCarb aims to produce column-averaged mixing ratios of $CO_2$, $CH_4$ and CO with accuracy per sample of $0.7\%$ ($\approx 2.7$ ppm), $1\%$ ($\approx 18$ ppb), and $10\%$ ($\approx 10$ ppb), respectively (Polonsky et al. (2014)). Geostationary orbit offers two main advantages over low Earth orbit (LEO). First, the signal-to-noise ratio (SNR) is proportional to the square root of the dwell time for detectors limited by photon shot noise, and geostationary orbits enables longer observation times, thereby increasing SNR. Second, due to the flexibility of the scanning mirrors, areas with high and uncertain anthropogenic emissions of $CO_2$, $CH_4$ and CO may be targeted with contiguous sampling, relatively small spatial footprints, and fine temporal resolution allowing for several observations per day on continental scales.

We are presented with the problem of finding an optimized scanning strategy for the GeoCarb satellite instrument. The underlying abstract mathematical problem related to optimizing the scanning pattern is the geometric set cover problem (Hetland (2014)). Given a finite set of points in space and a collection of subsets of those points, the objective is to find a minimal set of subsets whose union covers all the points in the space. The classical method for finding a solution to the geometric set cover problem is to employ a Greedy algorithm. Greedy algorithms incrementally choose optimal solutions based on the available information at a given time. In the context of the geometric set cover problem, the Greedy algorithm incrementally selects subsets that cover the most amount of uncovered points until all points are covered by the chosen subsets. Modifying the Greedy algorithm to optimize an objective function at each iteration is a common routine for finding geometric solutions to spatial problems with no known analytical solutions.

The task of determining the locations of new observation sites so that the total number of required sites to cover an area is minimal has been solved using IO algorithms (Rayner et al. (1996); Gloor et al. (2000); Patra and Maksyutov (2002)). Finding an optimized scanning strategy for GeoCarb is identical to an observation network optimization problem. Therefore, these IO algorithms were prospective candidates for application to GeoCarb. Our goal was to find a minimal covering set that translates to a scanning strategy that is operationally efficient and minimizes global measurement error for the GeoCarb instrument.

## 3 Methods

Translating the idea of the geometric set cover problem to GeoCarb's application, the collection of geometric subsets are 5-minute East-to-West scan blocks. The points in space to be covered are the North American and South American land masses

between $50°N$ and $50°S$ since this contains the regions relevant to the six science hypotheses mentioned in the introduction. Measurement errors are influenced by parameters that vary in space and time such as clouds, airmass, and solar zenith angle. Predicting cloud formation and quantifying the effects of clouds on measurement errors are active areas of research. For simplicity and computational efficiency, a cloud-free atmosphere is assumed in the simple instrument model. Surface albedo

is assumed to be constant within the span of a day. Due to time-dependency, solutions are in the form of ordered sets where the scan blocks are ordered by time of execution. These solutions are referred to in this paper as scanning strategies. With the simplifying assumptions making the problem computationally tractable and minimizing scan coverage over the ocean, a candidate set of 135 scan blocks is proposed in Fig. 1. This is a much larger candidate set than those of the network optimization studies that utilized evolutionary algorithms (Rayner et al. (1996); Gloor et al. (2000)). Therefore, the computationally efficient

IO algorithm, which is a modified Greedy algorithm, was implemented to select scan blocks that minimize our objective function at each increment in time.

### 3.1   Scan Blocks

The scanning region is discretized in the east-west direction assuming that GeoCarb will process commands in terms of 5-minute scan blocks. During the scan the instrument will step the slit from east to west within the scan block. Each slit observa-

tion is proposed to contain approximately 1000 individual soundings and is assumed to have a $\sim 9$ second integration time. The scanning region is further discretized in the north-south direction by scan blocks separated by $5°$ latitude increments. Potential scans that are primarily over the ocean are excluded, since measurements over the ocean are not a priority for the GeoCarb mission. The scan blocks are also restricted to land between $50°N$ and below $50°S$ as a hard constraint due to larger sensor viewing zenith angles at the higher latitudes, though this area still includes all regions relevant to the six science hypotheses

mentioned in the introduction. The resulting set of candidate scan blocks are shown in Fig. 1.

### 3.2   Science Operations Timeline

A goal of this study is to create a scanning strategy that views all land masses of interest at least once within the time window of usable daylight. To determine what time of day to begin the scanning process, Macapá, Brazil and Mexico City, Mexico were chosen as geographic reference points to determine the beginning and ending time, respectively, of the usable daylight time

frame. Macapá is located at $(0°, 50°W)$ at the mouth of the Amazon river and being on the equator gives us a consistent starting time relative to airmass factor (AF), a function of solar zenith angle (SZA) and the sensor viewing zenith angle (VZA), where $\mathrm{AF} = \frac{1}{\cos(\mathrm{SZA})} + \frac{1}{\cos(\mathrm{VZA})}$. Located at $(19.5°N, 99.25°W)$, Mexico City, Mexico is an ideal reference point to determine when the window of usable daylight ended because it is longitudinally centered in the North American land mass while being close enough to the equator for the calculated airmass factors to remain consistent through the winter months. The IO algorithm

calculates the starting time when Macapá first exceeds a starting threshold for AF and the ending time when Mexico City drops below an ending threshold for AF to determine when the usable daylight time window is over. As a result of parameter exploration experiments described in Sect. 3.6 , the suggested starting threshold is AF $= 2.6$ for the Summer Solstice and AF $= 2.7$ for the Autumn Equinox for minimum variance in predicted errors.

### 3.3 Uncertainty in Retrieved Gas Concentrations

Errors in retrieved gases arise from a result of numerous different sources, including imperfect radiometric calibration, errors in differential absorption spectroscopy, variations in the instrument line shape, and others. For simplicity, we assume that the errors in retrieved gases are arise from instrument noise as specified by a simple noise model arising from GeoCarb specific design parameters. The signal to noise ratio is then propagated through to uncertainty using a simple parameterization that was trained on retrieval results from simulated data.

The radiance observed by GeoCarb is an aggregate of insolation and atmospheric and land surface processes that absorb, reflect, and scatter photons. The impact of these processes is parameterized using a simple model, $I$, from Polonsky et al. (2014) that incorporates the effects of surface albedo and attenuation by aerosols over the sun-Earth-satellite path described by SZA and VZA:

$$I = F_{sun}\alpha\cos(SZA)e^{-AF\tau} \quad nW(cm^2 \, sr \, cm^{-1})^{-1} \tag{1}$$

where $F_{sun}$ is the band-specific solar irradiance, $\alpha$ is the band-specific surface albedo, and $\tau$ is the optical depth (OD) of atmospheric scatterers (e.g., aerosols, water). A cloud-free atmosphere is assumed for this simple model, whereas in the operational environment, clouds play a major role in retrieval quality due to poorly understood 3-D scattering effects. As can be readily verified, larger zenith angles lead to reduced signal for constant scatterer OD, as does smaller surface albedo. Note that $\tau$ is a quantity with significant spatial and temporal variability, as aerosol concentrations are modified by atmospheric dynamics, emissions, and chemistry. Typical values of $\tau$ in successful retrievals for OCO-2 are less than 0.6 for nadir soundings near the equator and decrease as AF increases. Similarly, surface albedo varies with land cover type on small spatial scales, and throughout the year with vegetation density. The OD term was set to $\tau = 0.3$ as it was previously found to be a reasonable estimate for a "clear" sky retrieval (Crisp et al. (2004), O'Dell et al. (2012)).

An important indicator of observation quality is the signal-to-noise ratio (SNR). In the case of GeoCarb, the signal is modeled as, $I$, and the instrument noise equivalent spectral radiance model, $N$, as

$$N(I) = \sqrt{N_0^2 + N_1 I} \quad nW(cm^2 \, sr \, cm^{-1})^{-1} \tag{2}$$

where $N_0$ and $N_1$ (in units $nW(cm^2 \, sr \, cm^{-1})^{-1}$) are parameters that empirically capture the effects of the instrument design (e.g., telescope length, detector noise) on overall instrument noise (O'Dell et al. (2012)). The Weak $CO_2$ band ($1.61\mu m$) specific constants that represent a signal-independent noise floor radiance, $N_0 = 0.1296$, and shot noise due to observed signal radiance, $N_1 = 0.00175$ are used in Eq. 2 to later calculate SNR. $N_0$ and $N_1$ are updated figures derived from the airborne trials with the Tropospheric Infrared Mapping Spectrometers (TIMS) by Lockheed Martin (Kumer et al. (2013)), and later revised in Polonsky et al. (2014). The SNR is then defined as $\frac{I}{N}$.

In O'Brien et al. (2016), the authors fitted an empirical model to predict the posterior errors, $\sigma$, estimated by the L2 retrieval algorithm as a function of the measurement SNR. In their case, $\sigma$ was derived from the L2 retrieval algorithm posterior

covariance given by

$$\hat{S} = (K^t S_\epsilon^{-1} K + S_a^{-1})^{-1} \tag{3}$$

where $S_\epsilon$ is the covariance of the instrument noise, $S_a$ is the covariance of the distribution about the prior state, and $K$ is the Jacobian of the transformation from states to measurements. This uncertainty represents the impacts of the noise on the fitted

spectra as well as nonlinearities in the radiative transfer model. It does not account for systematic errors that account from model deficiencies or instrument mis-characterization, which are beyond the scope of this work. O'Brien et al. (2016) found that the solid curves that best fit the posterior errors in the Weak $CO_2$ band were of the form $\sigma = \frac{a}{1+bx^c}$, where $x$ is the SNR and $a, b, c$ are real constants. For $CO_2$, $\sigma$ represents uncertainty in ppm. For a SNR of $x = 0$, the function takes its maximum value of $a$. Therefore, $a$ represents the prior uncertainty. With large values of $x$, the constant $c$ determines the rate of decay for

$\sigma$. Setting $a = 14$ ppm to express a conservative prior uncertainty on retrieved $CO_2$ and $c = 1$, the resulting empirical model was

$$\sigma = \frac{14}{1 + (0.0546)x}\text{ppm}. \tag{4}$$

The same model is used to connect SNR and uncertainty for evaluating scanning strategies later in this paper. For the purpose of our experiments, the distribution of $\sigma$ is treated as the metric against which a particular scanning sequence is evaluated.

## 3.4   Objective Function

Examining the definition of SNR, it is easy to see that $SNR \approx k\sqrt{I}$, where $k$ is a constant. Therefore it is sufficient to focus on maximizing $I$. Maximizing $I$ is equivalent to minimizing its multiplicative inverse, $\frac{1}{I}$. Therefore, an objective function was defined that is approximately proportional to $\frac{1}{I}$ on the parameters AF and surface albedo. In addition to minimizing SNR, two constraints were included in the objective function to prevent erratic scanning behavior. An overlap term, $\phi$, was introduced

to minimize repeated coverage of regions. A distance term, $\delta$, was also included to prevent erratic scanning behavior. $\delta$ is the shortest linear distance from the boundary of the last selected scan block to a candidate scan block. The objective function, c, to be minimized is given by

$$c(s,t) = \psi\left(1 + \frac{\phi + \delta^2}{\beta}\right), \tag{5}$$

where

$s =$ Candidate scan block.

        $t =$ Time.

        $\beta =$ Area of uncovered land mass covered by the candidate scan block

        $\phi =$ Area of overlapping coverage with selected blocks

        $\psi =$ Median of $e^{AF}\alpha^{-1}$ over the entire area of the candidate scan block

$\alpha =$ The surface albedo of a point within a scan block.

The terms $\phi$, $\beta$, and $\delta$ are illustrated in Fig. 2 for clarity. The median of $e^{AF}\alpha^{-1}$ is used because it is assumed that the distributions of airmass factor and surface albedo are non-Gaussian within the scan blocks due to the long viewing slit. The high variability of both parameters are described in Section 3.4.2.

### 3.4.1 Surface Albedo

The MCD43C3 Version 6 White Sky Albedo MODIS band 6 data set (Schaaf and Wang (2015)) was utilized for obtaining surface albedo, $\alpha$. The MODIS BRDF/Albedo product combines multiband, atmospherically corrected surface reflectance data from the MODIS and MISR instruments to fit a Bidirectional Reflectance Distribution Function (BRDF) in seven spectral bands at a 1 km spatial resolution on a 16-day cycle (Lucht et al. (2000)). The White Sky Albedo measure is a bihemispherical reflectance obtained by integrating the BRDF over all viewing and irradiance directions. These albedo measures are purely

properties of the surface, therefore they are compatible with any atmospheric specification to provide true surface albedo as an input to regional and global climate models. The native data was aggregated to the $0.5°$ spatial resolution, and interpolated in time to daily resolution.

### 3.4.2 Seasonal Variation of Parameters

Since AF is affected by the sun's position and albedo is affected by the density of vegetation, there are large seasonal variations

in both of these variables, shown in Figs. 3 and 4. However, there is little to no variation between day-to-day comparisons of these variables. It suffices then, and gives an added advantage of being computationally efficient, to calculate separate scanning strategies for each month rather than day.

## 3.5 Optimization Algorithms

The time-dependency of the scanning strategy requires the solutions to be represented as ordered scan blocks of the discretized

candidate set described in Sect. 3.1 and shown in Fig. 1. Therefore, the sum of permutations $\sum_{k=1}^{135} \frac{135!}{(135-k)!}$ gives approximately $7 \times 10^{230}$ possible solutions. Since it is computationally intractable to evaluate all possible solutions, a Greedy heuristic algorithm was employed to find a minimal covering set as a lower-bound estimate for the size of a solution set. The Greedy algorithm was then modified to an incremental optimization (IO) algorithm to find a scanning strategy optimizing for SNR.

### 3.5.1 Greedy Algorithm

Viewing the North American and South American land masses as a uniform space to be covered without considering any additional constraints, the problem is a geometric set cover problem where the goal is to find a minimal size covering set that we will call optimal. It is well-known that there are no known analytical solutions to the set cover problem, as it is one of Karp's 21 NP-Complete problems, and the optimization version is NP-Hard (Karp (1972)). However, there exists a heuristic method for finding a solution called the Greedy algorithm that selects the cover with the largest intersection with the uncovered

space recursively until the space is covered (Hetland (2014)). The pseudo-code of the Greedy routine is shown in Algorithm 1.

The Greedy algorithm is computationally efficient, but it is difficult to verify that the solution it finds is the optimal solution. The Greedy algorithm is suitable for the purpose of finding the smallest size scanning strategy because it reduces the set of candidate blocks at each iteration by removing the selected scan blocks to ensure that there are no repeated scan blocks in a solution. Running the Greedy heuristic with no objective function shows that the area of interest can be covered using 83 scan blocks. Therefore, this was taken as the lower bound of covering set size.

---

**Algorithm 1** Greedy Algorithm

---

$\mathbf{E} \leftarrow$ Space to be covered
$\mathbf{S} \leftarrow$ Set of scan blocks where $E \subseteq \bigcup s_i \in \mathbf{S}$
$\mathbf{I} \leftarrow \emptyset$
**while $\mathbf{E} \subsetneq \mathbf{I}$ do**
    Find $s^* \in \mathbf{S}$ such that $s^* \cap \mathbf{E}$ is maximal for all $s_i \in \mathbf{S}$
    Append $s^*$ to $\mathbf{I}$
    Remove $s^*$ from $\mathbf{S}$
**end while**

---

### 3.5.2 Incremental Optimization

The Greedy algorithm was modified to select the scan block that minimizes the objective function at each iteration to satisfy operational constraints. Presented in Patra and Maksyutov (2002), this modification to the Greedy algorithm is called an incremental optimization (IO) algorithm because its goal is to minimize the objective function at each increment of time to find the global optimum. Like the Greedy algorithm, IO has the advantage of being computationally inexpensive. However, it may find local optima only and produce sub-optimal solutions depending on the nature of the problem. Usually to avoid this issue, small perturbations are introduced at each increment, such as is done in evolutionary algorithms (e.g., simulated annealing and genetic algorithm). It has been shown that IO yields results that are nearly as good as evolutionary algorithms while using a fraction of the computational power (Nickless et al. (2018)).

For GeoCarb's application, we were looking at the global distribution of errors, $\sigma$, and therefore were not concerned about local optima. An additional constraint was added that required the algorithm to cover South America before switching to North America to further prevent erratic scanning behavior. The pseudo-code of the IO algorithm is shown in Algorithm 2.

### 3.6 Parameter Exploration

The IO algorithm calculates a scanning start time from a specified starting AF threshold, described in Sect. 3.2, but it was unknown what overall effect the AF threshold had on the overall performance of the resulting scanning strategy. In the objective function (Eq. (5)), the overlap and distance terms had equal weighting and different weightings were tested to understand their effects as well. A Monte Carlo experiment was performed to determine the distribution of sample error statistics across a range of possible starting AF thresholds and weights for overlap and distance. The effects of different weightings of the distance and

**Algorithm 2** Incremental Optimization Algorithm

---

$\mathbf{E} \leftarrow$ North American and South American land masses between $50°N$ and $50°S$

$\mathbf{S} \leftarrow$ Set of 5-minute east-to-west scan blocks where $E \subseteq \bigcup s_i \in \mathbf{S}$

$\mathbf{I} \leftarrow \emptyset$

$\mathbf{C} \leftarrow$ Objective Function

**while** $\mathbf{E} \subsetneq \mathbf{I}$ **do**

    Find $s^* \in \mathbf{S}$ such that $\mathbf{C}(s^*)$ is the minimum of the set $\{\mathbf{C}(s_i) : s_i \in \mathbf{S}\}$

    Append $s^*$ to $\mathbf{I}$

    Remove $s^*$ from $\mathbf{S}$

**end while**

---

overlap terms on the global distribution of errors were investigated specifically by adding $(w_o, w_d)$ constant weight terms to Eq. (5) as new input parameters resulting in

$$c(s, t, w_o, w_d) = \psi(1 + \frac{w_o \phi + w_d \delta^2}{\beta}). \tag{6}$$

Applying Equation (6) to the algorithm gives the operator three inputs to specify, $w_o$, $w_d$, and the starting AF threshold. For

both $w_o$ and $w_d$, 1000 weights were randomly sampled from a uniform distribution between 0 and 10. This process was repeated for the Summer Solstice and Autumn equinox for starting AF thresholds starting from 2.5 increasing by 0.1 to 3.5 for a total of 22,000 experiments. For these experiments, the contiguous land masses of North and South America were scanned with equal importance. The minimum variance of predicted errors with respect to starting AF threshold occured at 2.6 for the Summer Solstice and 2.7 for the Autumn Equinox , shown in Fig. 6. Both distributions of median and variance of errors averaged a

0.01 ppm spread over all values of $w_o$ and $w_d$ tested. Therefore, it was concluded that the effects of different weightings of the distance and overlap terms were negligible on the overall aggregate error and weighting terms were excluded from the objective function. A sensitivity analysis was also done to quantify the effects of these results and can be found in Appendix A.

### 3.7   Evaluating the Optimized Scanning Strategy

For evaluating the performance of an algorithm-selected scanning strategy, the empirical distributions of error, $\sigma$ (Eq. (4)), were compared between the optimized strategy and a baseline scanning strategy proposed in Moore et al. (2018). An example of the two strategies are shown in Fig. 5. The baseline strategy tracks the sun's path from East to West and covers the entire area of interest in five coherent regions in the order of Tropical South America East, Tropical South America West, Temperate South America, Tropical North America, and Temperate North America. The same scanning start times used by the IO algorithm are

used for evaluating the performance of the baseline strategy. The times are calculated by the algorithm, based on a starting AF threshold supplied by the user, were 1230 UTC for the Autumn Equinox with a starting AF threshold of 2.6 and 1315 UTC for the Summer Solstice with a starting AF threshold of 2.7.

In practice, a post-processing filter (PPF) is applied to retrieved satellite data and the data is marked with a quality flag to notify the end-user of its overall usefulness. For this study, a threshold of 100 on the SNR is used as our PPF to determine a "usable" sounding. This threshold limits the predicted error to a maximum of $\sim 2$ ppm, (Eq. (4)), and is within the accuracy per sample performance requirements laid out in Polonsky et al. (2014).

## 4   Experiment 1 – Equal importance for all land masses

In the first experiment, all contiguous land masses of North and South America were scanned with equal importance. Based on the parameter exploration results, simulations were performed for the Summer Solstice with a starting AF threshold of 2.6 and the Autumn Equinox with a starting AF threshold of 2.7. The algorithm-selected scanning strategies consistently matched or exceeded the performance of the baseline scanning pattern, shown in Figs. 8 and 9. The region where the most significant improvement is seen is in the Amazon during the Autumn Equinox, refer to Fig. 10. After applying the PPF to the simulation results, it was clear that the greatest performance increase from the baseline strategy was in usable soundings. During the Summer Solstice, the algorithm-selected strategy yielded $\sim$3.79 million usable soundings versus $\sim$3.02 million usable from soundings the baseline strategy. During the Autumn Equinox, the algorithm-selected strategy yielded $\sim$4.31 million usable soundings versus $\sim$3.04 million usable soundings from the baseline.

Part of the increase in usable soundings can be attributed to the optimized strategy following the coastline better, which results in less scans over the ocean and more overlapping scans, refer to Fig 5. However, the comparison of SZA and AF between the baseline and algorithm-chosen strategy shows that the algorithm also selects more scan blocks with low SZA and low AF, refer to Fig (11). It is important to note that these figures are results from simulations done in the cloud-free environment of the model. Realistically, there is a high probability that parts of the scanning slit will include cloudy scenes, we expect the yield of usable soundings to be significantly less during operations, but those effects will be seen similarly in both the baseline and optimized strategies.

## 5   Experiment 2 – City campaign

A major advantage of having a geostationary platform is the flexibility to scan areas of high interest at times of optimal observing conditions. In this section, a "temporary campaign" mode is demonstrated where GeoCarb observes the ten most populous cities in North and South America as areas of high interest, which are New York, Chicago, Los Angeles, Dallas Fort-Worth, Mexico City, Bogota, Sao Paolo, Rio de Janeiro, Lima, and Buenos Aires. The demonstration is done for the Autumn Equinox with a starting AF threshold of 2.7. The areas of interest are given higher weighting in the algorithm through a modified version of Eq. (5). The performance of the resulting optimized strategies are compared to the baseline strategy, both globally and for the ten cities of interest.

## 5.1 Modified Objective Function

To give these areas of interest a higher weight, a time-dependent scaling factor was added to the term $\psi$ in the objective function (Eq. (5)) for scan blocks containing these cities, refer to Fig. 12. The scaling factor is defined as, $e^{b-c}$, where $b$ is the AF of a point with respect to time and $c = a + e^{2-a}$, where $a$ is the daily minimum AF of the point. The term $c$ acts as a threshold for the selection of the scanning block. While $b$ is greater than $c$, the scaling factor penalizes the objective function by giving it a larger value, which tells the algorithm to wait on selecting the block until it is reasonably close to its minimum AF. Once $b$ becomes less than $c$, the scaling factor scales down the value of the objective function to make the algorithm select the scan block as soon as possible. Figure 13 shows the scaling factor for a point with a minimum daily AF of 2. Table 1 shows the relationship between minimum daily AF and the scaling factor threshold for a sample of minimum AFs. The modified objective function is

$$c(s,t) = \tilde{\psi}(1 + \frac{\phi + \delta^2}{\beta}), \tag{7}$$

where $\tilde{\psi}$ is the median of $e^{b-c}e^{AF}\alpha^{-1}$ over the entire area of a candidate scan block containing a city of interest.

## 5.2 Predicted errors

The addition of the scaling factor only affects the candidate scan blocks that contain a city of interest. Hence, there should be no significant degradation in the overall performance of the optimized scanning strategy. Fig. 14 shows that there is still a significant increase in usable soundings, $\sim 3.97$ million versus $\sim 3.03$ million globally.

Looking at only observations over the ten cities, the optimized scanning strategy shows an increase of $\sim 2000$ usable soundings over the baseline strategy, refer to Fig. 15. Shown in Fig. 16, the baseline strategy's city observations have a higher concentration of low SZA soundings, but the optimized strategy's city observations have a higher concentration of low AF soundings.

## 6 Conclusions

We illustrate an efficient, offline technique that creates a geostationary scanning strategy that minimizes overall predicted measurement error. Applied in a simplified instrument model of GeoCarb, the IO routine gives us an optimized scanning strategy that performs better than the baseline scanning strategy relative to the global distribution of error and number of usable soundings. In Section 4, we showed that the incremental optimization of SNR with respect to the stationary physical processes, AF and albedo, results in an overall performance increase with the region of greatest performance increase seen in the Amazon (Fig. 9). We have also shown in Sect. 5 that the IO routine can be easily modified for a temporary campaign mode that focuses on the ten most populated cities of North and South America. Other examples of possible scenarios for temporary campaigns are wildfires, droughts, and volcanic eruptions.

At the moment, our model does not take into account the effect of clouds on retrieval quality. It is known that clouds play a significant role in scattering effects and influences $\tau$ in the calculation of radiance (Eq. (1)), but quantifying these effects is an

active area of research. In a case study including clouds and aerosols in the atmosphere performed by Polonsky et al. (2014), the authors found that the number of usable soundings passing their post-processing filter (PPF) of aerosol optical depth (AOD) $< 0.1$ was between 8.1% to 20% of total simulated soundings. We believe that an AOD threshold of 0.1 is too strict for the clear sky atmosphere used in our simulations, therefore the threshold was relaxed to 0.3 to capture a conservative estimate of usable

soundings as previously done by O'Dell et al. (2012) and Rayner et al. (2014). O'Dell et al. (2012) found that 22% of their simulated observations were classified correctly as "clear" when they used an AOD threshold of 0.3. Because we set $\tau = 0.3$ in our calculation of radiance (Eq. (1)), our estimate is that the true number of usable soundings will be around 20% of our simulated usable soundings in Sect. 4. Going forward, the incorporation of cloud products from CALIPSO will be investigated to better simulate operational conditions and produce more robust estimates of usable soundings.

The SNR-optimized scanning strategy outperforms the proposed strategy for the GeoCarb scientific observation plan. An empirical model that calculates predicted $CO_2$ retrieval uncertainty, $\sigma$, as a function of SNR was used to evaluate the performance of algorithm-selected strategies. The optimized scanning strategies consistently matched or exceeded the predicted performance of the proposed scanning strategy pattern with respect to aggregate distribution of $\sigma$. When a simple post-processing filter (PPF) of SNR > 100 was applied to determine what constituted a usable sounding, the optimized strategies yielded a

$\sim 18\%$ increase of usable soundings during the Summer and a $\sim 41\%$ increase during the Autumn over the proposed scanning strategy.

*Data availability.* The MCD43C3 MODIS BRDF/Albedo data was retrieved from the online Data Pool, courtesy of the NASA EOSDIS Land Processes Distributed Active Archive Center (LP DAAC), USGS/Earth Resources Observation and Science (EROS) Center, Sioux Falls, South Dakota, https://lpdaac.usgs.gov/.

**Appendix A: Sensitivity Analysis**

To quantify the algorithm's sensitivity to input parameters, the method of standardized regression coefficients (SRC) was utilized (Helton et al. (2006)). SRCs are the regression coefficients of a linear model fitted to the standardized dependent variable, $Y_Z = \frac{Y - \bar{Y}}{\sigma_Y}$, using standardized independent variables, $X_Z = \frac{X - \bar{X}}{\sigma_X}$. The dependent variable in this case is the predicted error and the independent variables are $w_o$, $w_d$, and the starting AF threshold. The standardization of variables allows for measuring

the effect of the input parameters without their dependency on units (i.e., ppm). The coefficient of determination, $R^2$, of the SRC model tells us how much of the variability in the sample statistics is explained by the SRC model. $R^2$ is defined as the

Modeled Sum of Squares (MSS) divided by the Total Sum of Squares (TSS), where

$$\text{MSS} = \sum_{i=1}^{n} (\hat{Y}_i - \bar{Y})^2$$

$$\text{TSS} = \sum_{i=1}^{n} (Y_i - \bar{Y})^2$$

$$R^2 = \frac{\text{MSS}}{\text{TSS}}$$

and $\hat{Y}$ = model predicted values, $\bar{Y}$ = mean error, $Y$ = observed values, $n$ = number of observations. The method of SRC was chosen for the sensitivity analysis by convenience of readily available simulation data from the parameter exploration experiment.

The SRCs show that both the median and variance of the global error are found to be sensitive to starting AF thresholds as seen in Figs. 6 and 7, and Tables 2 and 3. This sensitivity was expected considering that airmass factors depend on time and

play a large role in the calculation of radiance (Eq. (1)). The starting AF thresholds affect the scanning strategy as a whole by shifting the scanning time frame. Because SRCs determine the effect of the input parameters in the presence of others, the SRCs fitted to a linear model of predicted error with respect to $w_o$ and $w_d$ were also analyzed within the Monte Carlo samples of starting AF threshold equal to 2.7 for the Autumn Equinox and starting AF threshold equal to 2.6 for the Summer Solstice.

Within the specified starting AF threshold of 2.7 for the Autumn Equinox, moderate effects of the weights were found on the

sample global error distribution. The values in Table 4 show that the SRC model explains approximately half of the variability in median of global error distributions, $R^2 = 0.552$, and the parameter with the largest effect on the variance is the distance, $\delta$. With respect to variance of global error distributions, the SRC model explains less than half of the variability with $R^2 = 0.384$. Again, the parameter with the largest effect is the distance term.

Within the specified starting AF threshold of 2.6 for the Summer Solstice, the effects of the weights on the sample global

error distribution are small. The SRC model explains around a quarter of the variability in median of global error distributions, $R^2 = 0.242$, and $\sim 15\%$ of the variability with $R^2 = 0.148$, shown in Table 5. The parameter with the largest effect is the overlap term for both variance and median of error distributions.

$R^2$ values less than 0.7 signify small sensitivity to the independent variables or a nonlinear relationship between the independent and dependent variables. Visual analysis of the scatter plots of the distributions of sample statistics versus weights

(Fig. A1) does not imply a nonlinear relationship between the weights and sample statistics. It is important to note as well, that the non-standardized sensitivity of predicted errors with respect to $w_o, w_d$ results in a spread of 0.01 ppm in the overall performance of an algorithm-selected scanning strategy. We conclude that the weighting terms contribute negligible effects to the algorithm's performance.

*Author contributions.* SC conceptualized the goals of this project. JN and SC developed the methodology together. BM provided oversight

and guidance. JN prepared the manuscript, developed the algorithm code, model code, and executed the simulations.

*Competing interests.* The authors report no competing interests.

*Acknowledgements.* Some of the computing for this project was performed at the OU Supercomputing Center for Education & Research (OSCER) at the University of Oklahoma (OU). This work was supported by NASA award No. 80LARC17C0001.

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

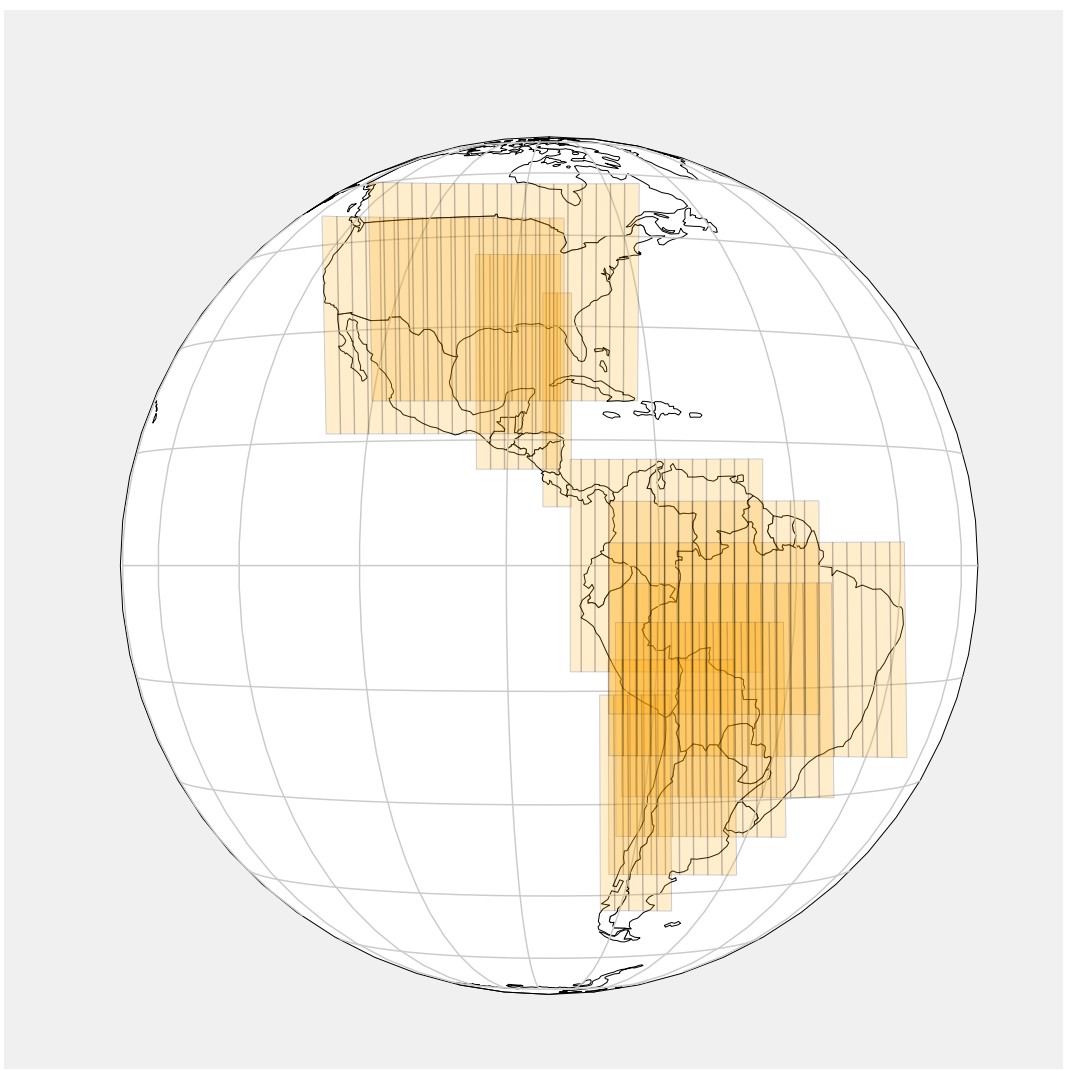

**Figure 1.** Candidate Scan Blocks.

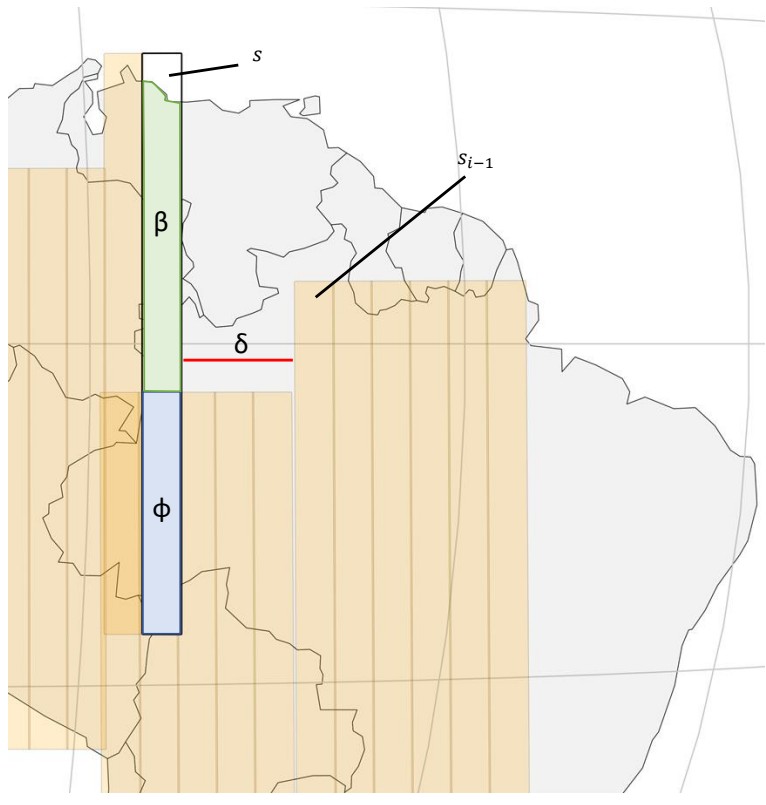

**Figure 2.** A diagram explaining the objective function, c(s, t), used in the IO routine. The block labeled, $s_{i-1}$, is the last selected block and the block labeled, $s$, is the block for which c(s, t) is being calculated.

| $a$ | $c$ | $c - a$ |
|------|-------|---------|
| 2.0 | 3.0 | 1 |
| 2.3 | 3.041 | .741 |
| 2.5 | 3.107 | .607 |
| 2.7 | 3.197 | .497 |
| 3.0 | 3.36 | .36 |
| 4.0 | 4.135 | .135 |
| 4.95 | 5.0 | .05 |

**Table 1.** This table shows for a sample of daily minimum AFs the distance relationship between the daily minimum AF of an observed point, $a$, and the scaling factor threshold, $c$, used in the modified objective function (Eq. (7)).

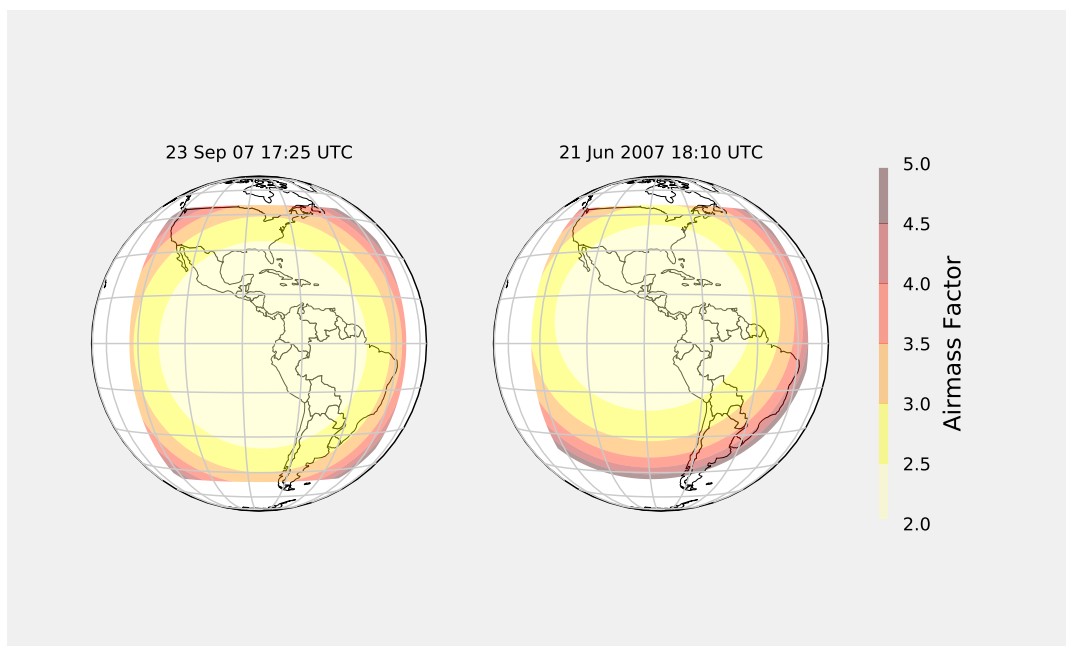

**Figure 3.** Comparing airmass factors in September (left) and June (right).

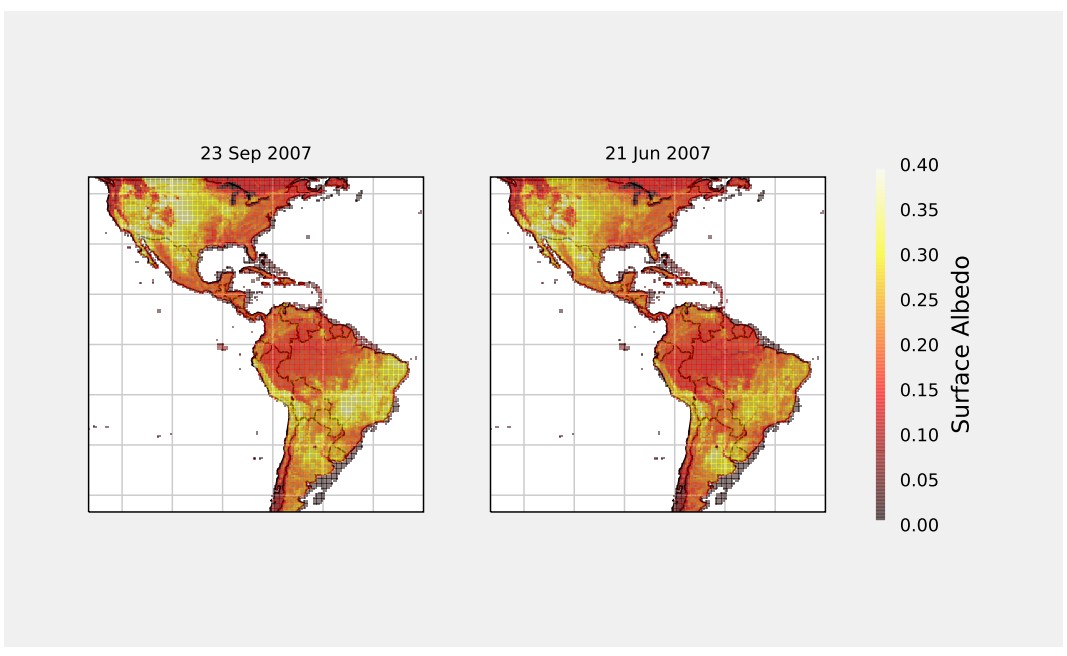

**Figure 4.** Comparing Surface Albedo during the Autumn Equinox (left) and the Summer Solstice (right).

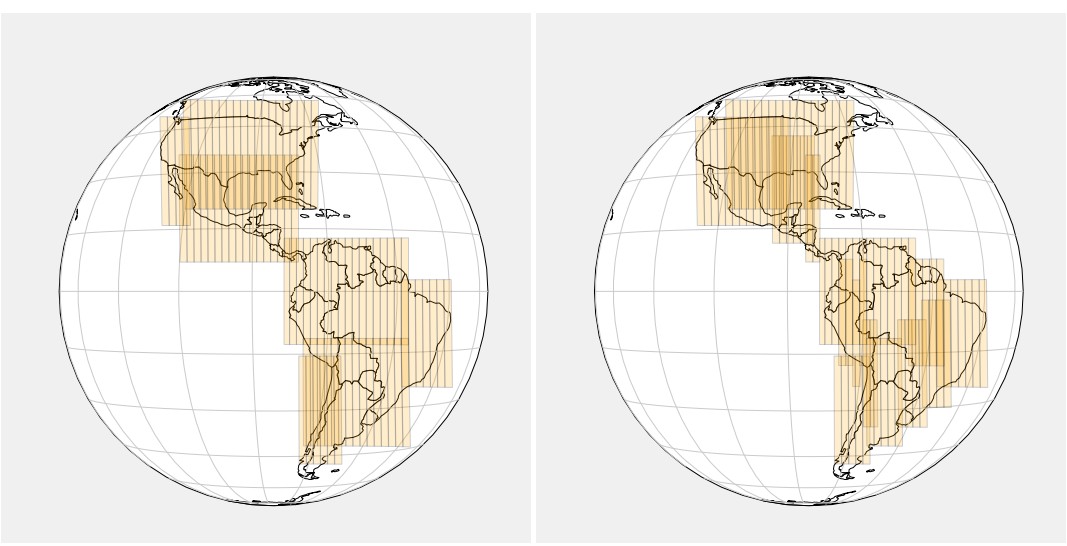

**Figure 5.** The baseline strategy (left) compared to an algorithm-selected strategy (right).

| Standardized Regression Coefficients, Summer Solstice | | | |
|---|---|---|---|
| Input Parameters | Variance | Median | Expected Usable Observations |
| $R^2$ | 0.939 | 0.935 | 0.311 |
| Starting Threshold | 0.9679 | 0.9618 | -0.2769 |
| $w_o$ | 0.0615 | -0.0621 | -0.0634 |
| $w_d$ | -0.0040 | -0.0716 | 0.4839 |

**Table 2.** The SRCs show that the variance and median of global error distributions are sensitive to starting AF thresholds.

| Standardized Regression Coefficients, Autumn Equinox | | | |
|---|---|---|---|
| Input Parameters | Variance | Median | Expected Usable Observations |
| $R^2$ | 0.646 | 0.977 | 0.208 |
| Starting Threshold | 0.7997 | 0.9770 | -0.4473 |
| $w_o$ | -0.0163 | 0.0056 | -0.0579 |
| $w_d$ | 0.0757 | 0.1455 | 0.0772 |

**Table 3.** The SRCs show that the median of global error distributions is sensitive to starting AF thresholds. The low $R^2$ value for variance indicates that there may be a nonlinear relationship between variance and starting AF threshold. Fig. 6 shows that this is the case.

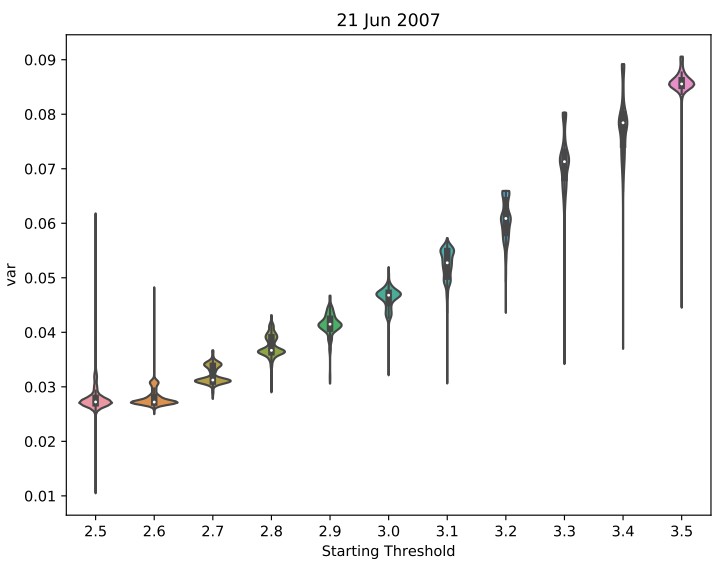

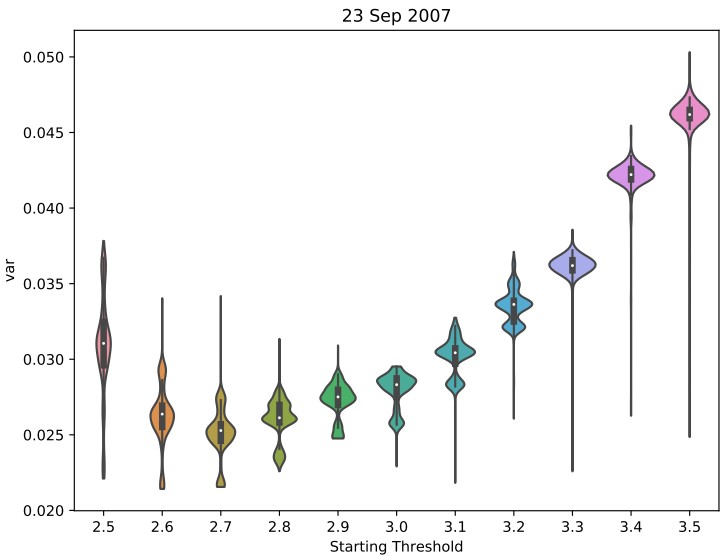

**Figure 6.** Violin plots show the effect of starting threshold on variance of errors: Summer Solstice (top) and Autumn Equinox (bottom).

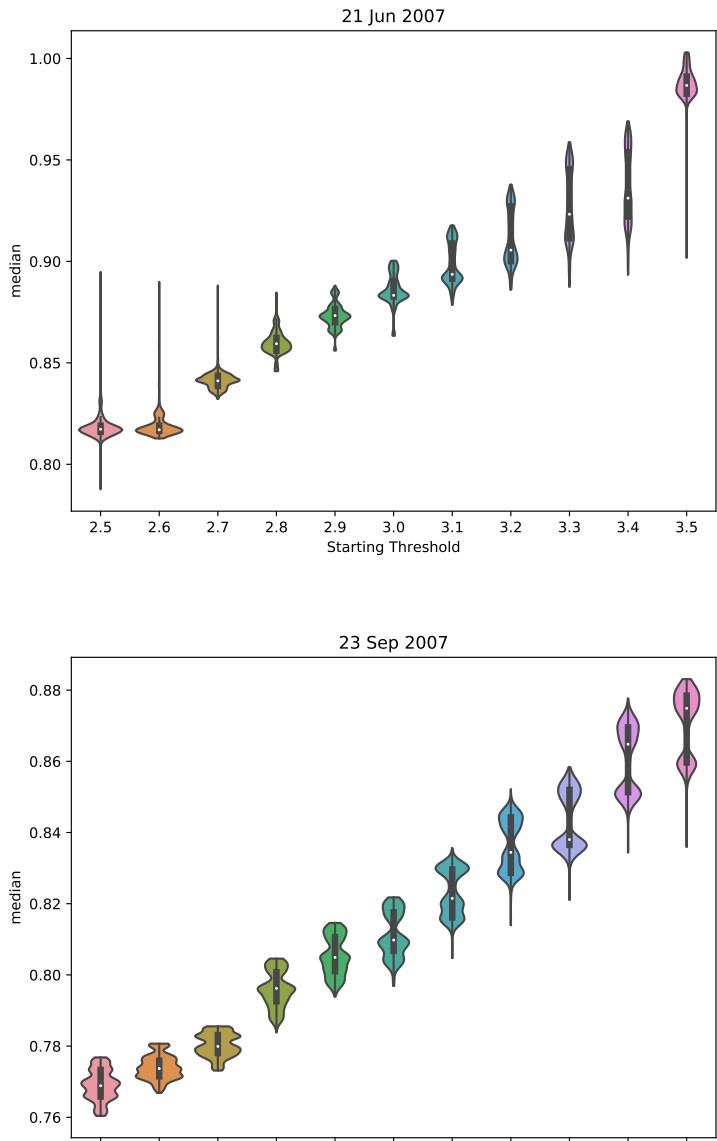

**Figure 7.** Violin plots show the effect of starting threshold on error distribution medians: Summer Solstice (top) and Autumn Equinox (bottom).

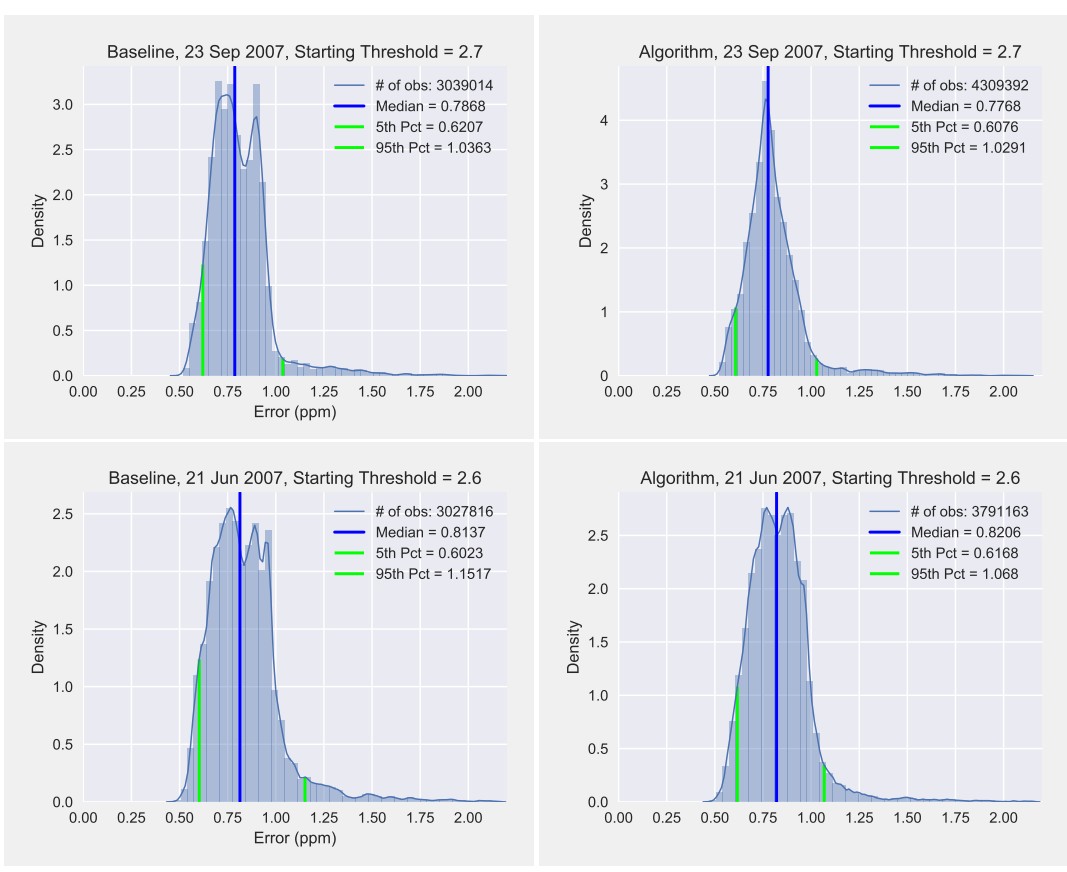

**Figure 8.** Global error distribution, baseline strategy (left) and algorithm-selected strategy (right).

| SRC for Starting Threshold = 2.7, Autumn Equinox | | | |
|---|---|---|---|
| Input Parameters | Variance | Median | Expected Usable Observations |
| $R^2$ | 0.384 | 0.552 | 0.497 |
| $w_o$ | -0.3051 | 0.2502 | -0.2310 |
| $w_d$ | 0.5450 | 0.6950 | 0.6702 |

**Table 4.** The SRCs show that the median and variance of global error distributions are not senstive to different weighting of the distance and overlap terms. $R^2 < 0.7$ usually signifies insensitivity to independent variables.

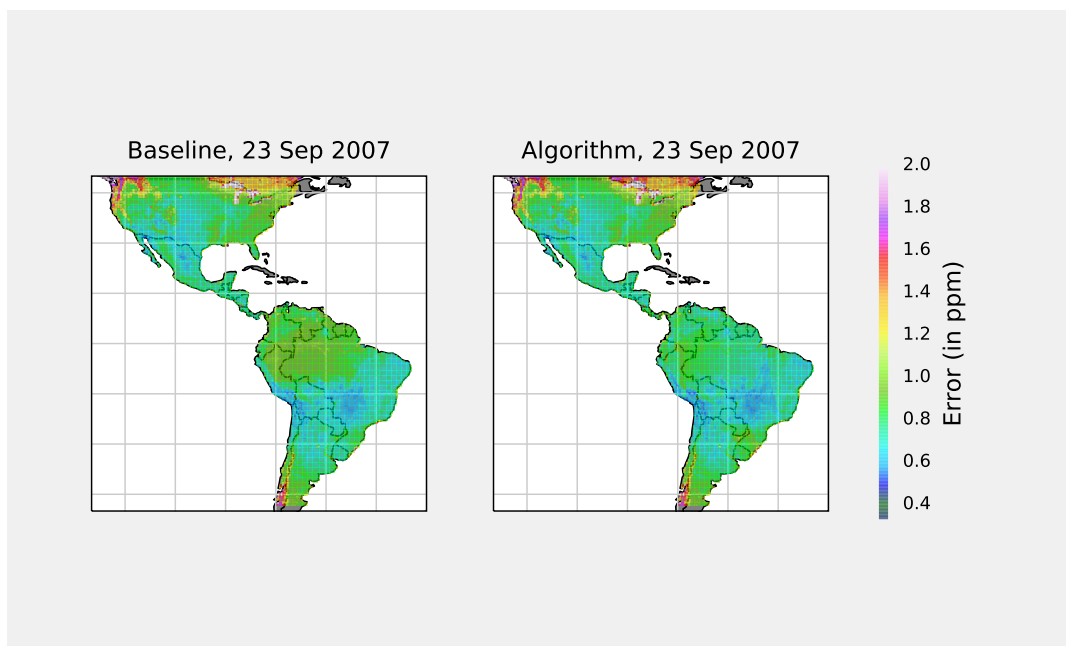

**Figure 9.** Comparing the algorithm-selected strategy (right) to the baseline strategy (left).

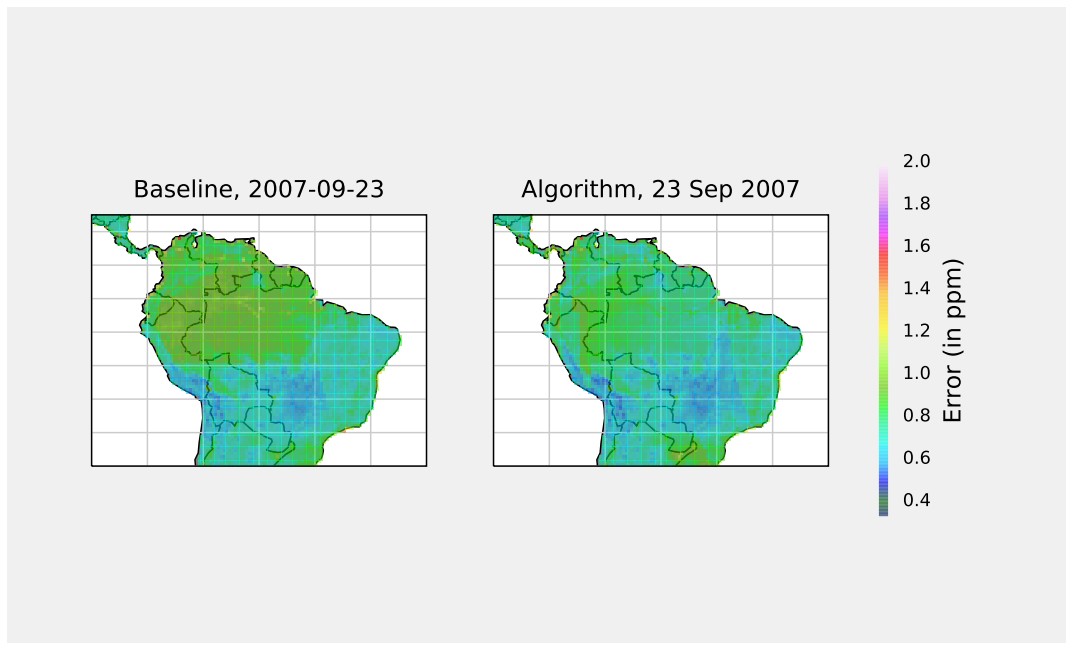

**Figure 10.** There is a significant improvement in predicted errors over the Amazon for the Autumn Equinox.

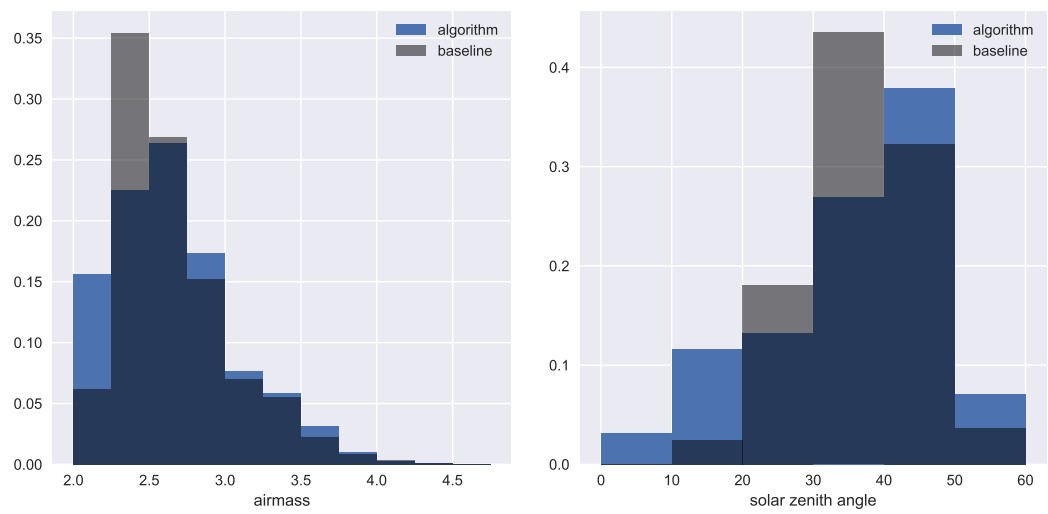

**Figure 11.** The histograms show that the algorithm selects more scan blocks with low AF (left) and low SZA (right) than the baseline strategy.

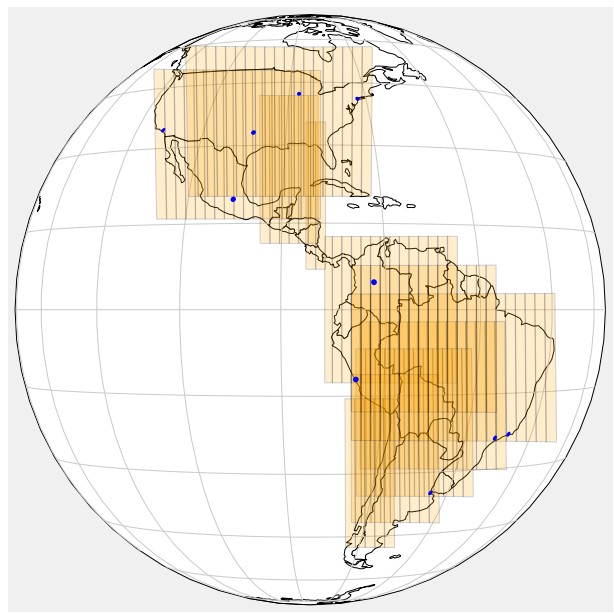

**Figure 12.** Scan blocks containing the ten most populated cities are given higher weighting in city campaign mode.

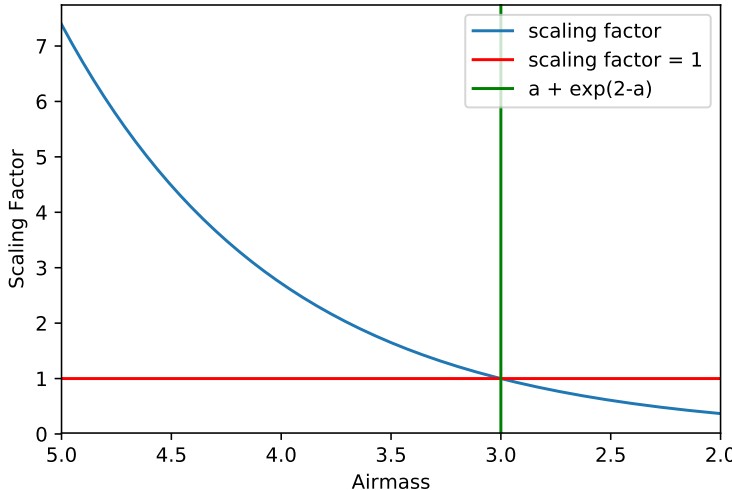

**Figure 13.** Scaling Factor for an observation point with minimum daily airmass factor, $a = 2$.

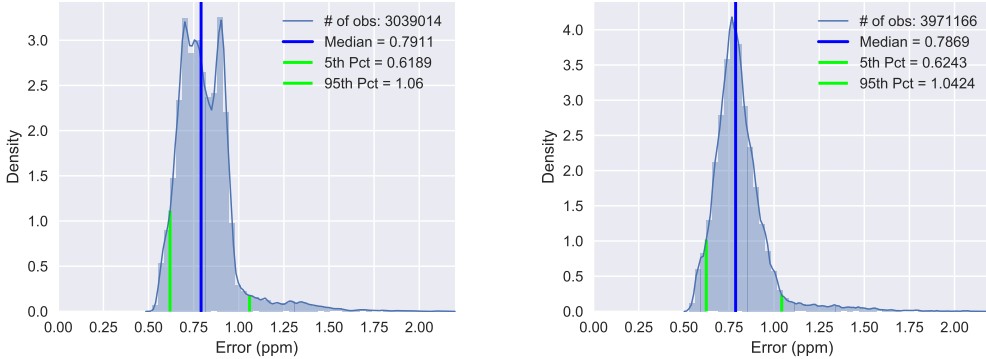

**Figure 14.** Compared to the baseline strategy (left), the overall performance of the algorithm-selected strategy (right) is not significantly degraded in the city campaign mode.

| SRC for Starting Threshold = 2.6, Summer Solstice | | | |
|---|---|---|---|
| Input Parameters | Variance | Median | Expected Usable Observations |
| $R^2$ | 0.148 | 0.242 | 0.284 |
| $w_o$ | -0.3481 | -0.4833 | -0.3911 |
| $w_d$ | -0.1717 | -0.1064 | 0.3519 |

**Table 5.** The SRCs show that the median and variance of global error distributions are not sensitive to different weighting of the distance and overlap terms. $R^2 < 0.7$ usually signifies insensitivity to independent variables.

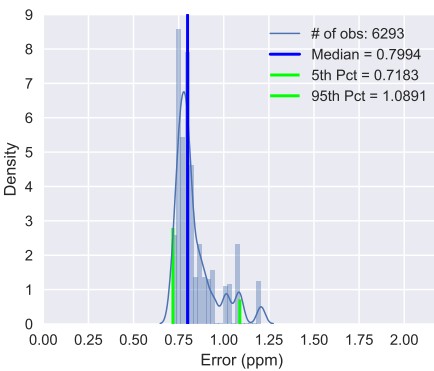
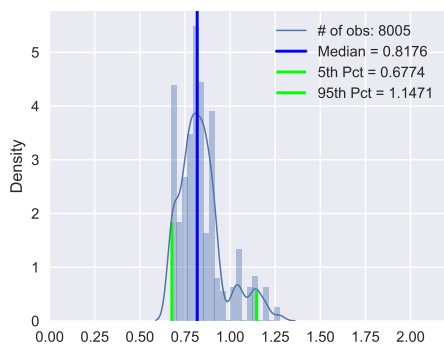

**Figure 15.** Compared to the baseline strategy (left), the algorithm-selected strategy in city campaign mode (right) sees an increase of approx. 2000 usable soundings over the ten most populated cities.

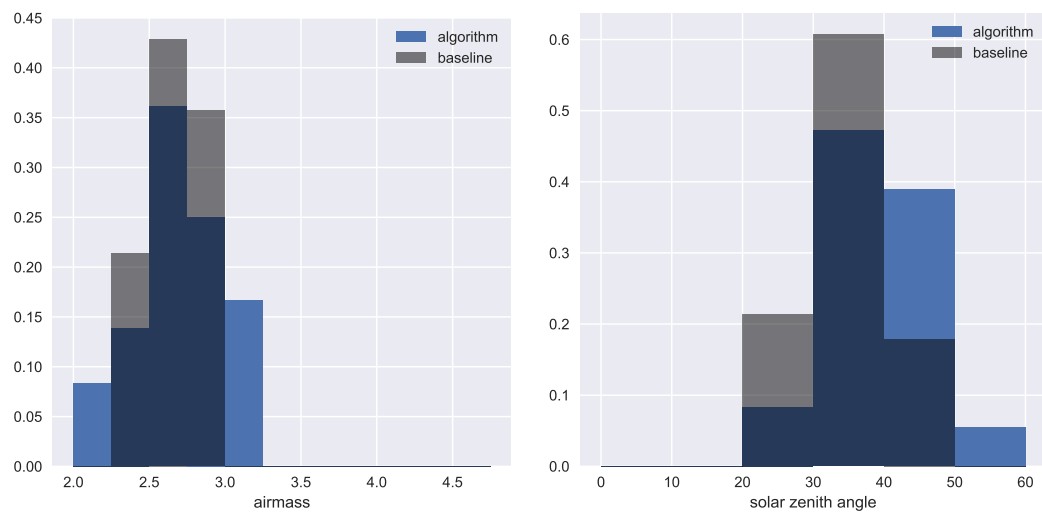

**Figure 16.** In the city campaign mode, the histograms show that the algorithm-selected strategy has more observations with low AF (left), but the baseline strategy's observations have lower SZA (right).

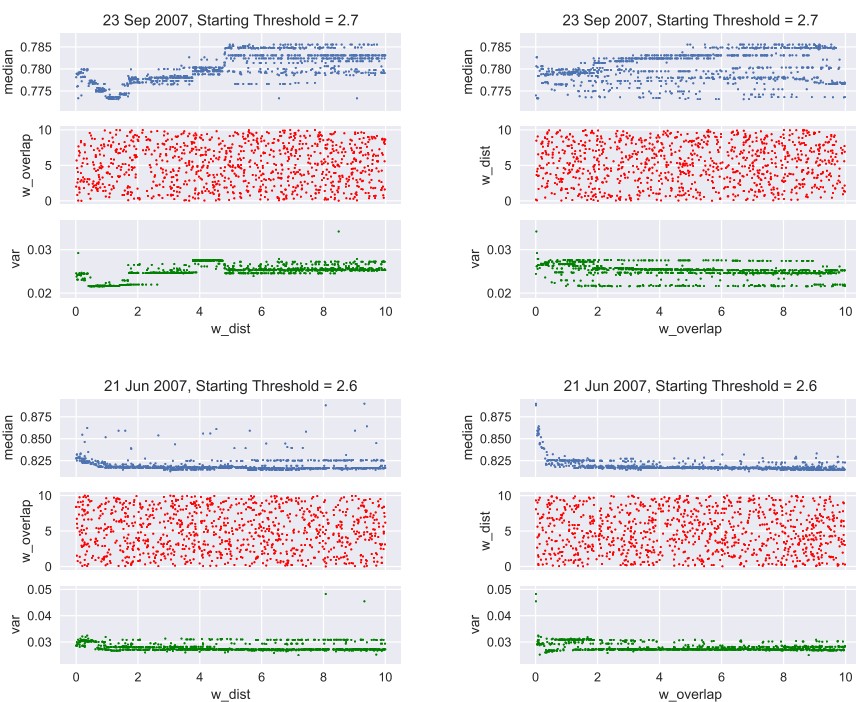

**Figure A1.** Scatter plots do not indicate a nonlinear relationship between weights and sample statistics. $w_o$ and $w_d$ are indicted as w_dist and w_overlap in the x-axis.