# Peer review of "An SNR-Optimized Scanning Strategy for the Geostationary Carbon Cycle Observatory (GeoCarb) Instrument"

_Atmospheric Measurement Techniques, 2018_

## Referee Comment (RC1) · Anonymous Referee #1 · 17 Dec 2018

This paper develops a scanning strategy for the upcoming GeoCarb mission from geostationary orbit, using an optimization algorithm to establish the greatest return in terms of soundings that exceed an unspecified minimum threshold in signal-to-noise ratio (SNR).

The development of the approach appears logical and seems to give reasonable results. To me, though, it feels like a starting point for a more detailed treatment. It treats the land masses of the Americas as a "uniform space," treating all points as equally important. While it is crucial to obtain global coverage (over the viewing area of the satellite), it seems to me that not all locations are equivalent in terms of monitoring

greenhouse gas emissions. There are presumably hot spots of industrial activity that could benefit from closer scrutiny. The measurements do no extend to high enough latitude to capture emissions from the Alberta oil sands, but you will measure over the U.S. oil shale deposits in Colorado, Utah, and Wyoming, for example.

I say this because one of the purported benefits of having the mission on a geostationary platform was that, according to the manuscript, "areas with high and uncertain anthropogenic emissions of CO2, CH4 and CO may be targeted with contiguous sampling," but this benefit is not exploited with the proposed scanning strategy. One would need to attribute increased weight to the hot spots to properly shade the coverage. The authors do mention the notion of adjusting the coverage to study events such as volcanoes and large wildfires, but that is a separate notion, a temporary campaign mode rather than a regular coverage strategy.

Another benefit mentioned for the geostationary platform was that you could improve the signal by increasing the dwell time for a measurement. I do not know if there is some constraint that would require every scan to be identical. If one could select the measurement dwell time employed in each individual scan, for example, one could improve the results in regions where a large percentage of the soundings would otherwise fall below the SNR threshold and therefore be tagged as unusable. In this scenario, scan time would become an additional parameter to include in the optimization.

The authors indicate that no results above the oceans are possible due to low signal. Is there no possibility of making use of ocean glint, as the OCO-2 mission does? This would obviously only work at certain times, when the conditions were such that sunlight reflected off the ocean at the same angle that the instrument was viewing the surface, but it would expand the coverage.

Judging from Figure 4, the gain in "usable soundings" relative to the baseline approach appears to be strongly related to a reduced number of measurements over the oceans. The expected improvement in errors over the Amazon is presumably related to the

increased number of overlapping scans in that region.

Note that no mention was made of what constituted a usable sounding. The last sentence relates the gain in soundings with SNR > 100, but it is not clear if that was the threshold employed for the determination of whether a particular measurement was usable. Based on the discussion on page 4, the calculated SNR was associated with the O2 A-band measurements. That means you assumed the SNR for the measurements of the CO2 bands was higher, or at the very least comparable?

In the text (page 8, lines 9-10), the statement is made that Figure 6 showed the minimum error distribution medians and variances occur where both weights are equal to 1. For me, that does not seem like an obvious conclusion to draw from the figure. It certainly seems true for the upper left panel, and maybe for the median in the upper right panel, but unless I misinterpret what is being said, I do not see it for the other plots.

Minor comments

Page 2, line 2: the acronym "FoV" was defined but not used again, so there is no need to define the acronym

Page 8, line 9: "Figure 6" should be "Fig. 6"

Page 8, line 17: "Fig. 7 and 8" should probably be "Figs. 7 and 8"

Page 9, line 9: . . . mean Error . . . >Why is "Error" capitalized?

Page 9, line 13: "Fig. 5 and 10" should probably be "Figs. 5 and 10"

Page 10, lines 4-5: "We also found that by optimizing for the global distribution of error, we obtained an improvement in regional errors as well, seen in Fig. 8" >This is not true for all regions, maybe an "overall improvement"

Page 10, line 13: the acronym "AOD" is not defined

Figure 2: The caption claims that the plots relate to June (on the left) and December (on the right). The titles on the plots suggest they relate to September (on the left) and June (on the right).

Figure 6: The variables w_dist and w_overlap are used to label the plots rather than w_d and w_o, the variables employed in the text.

In Figures 8 and 9, it looks like there are no results over Cuba (greyed out?), even though that region appears to be within the scan range.

---

## Referee Comment (RC2) · Anonymous Referee #2 · 13 Feb 2019

General Comments

The manuscript deals with the optimization of geographic coverage, which is a problem of interest for geostationary satellite remote sensing. The approach presented in the manuscript is new and deserves publication in AMT. However, the approach and its underlying assumptions need to be better explained and the discussion of the results needs to be made clearer. The manuscript needs to be revised addressing the issues identified below.

Specific Comments

1. The proposed scheme aims at enhancing the yield and quality of a geostationary

[Figure]

CO2 observation system by optimizing the scanning strategy with a focus on the Signal to Noise Ratio (SNR). Many other parameters that are expected to drive the yield and quality of such observations are not taken into account. Degraded CO2 product quality is expected not only in cases with low SNR, but also in many other conditions e.g. when viewing geometries are slant, when target air masses contain clouds or aerosol, and when clouds cover parts of the field of view thus increasing the risk of spatial stray light. The choice of focusing on SNR needs to be justified, and the approach regarding other potential drivers need to be explained and motivated.

2. The link between radiometric noise and the total CO2 uncertainty need to be discussed in more detail (beyond reference to O'Brian, 2016 and Eq. 3). The main contributors to the CO2 product uncertainty budget need to be discussed, and it needs to be explained why the optimization is driven by the random radiometric error.

3. The objective function given (Eq. 5) minimized in the optimization scheme seems incomplete. The SNR depends on the radiance signal level (Eq. 2) hence also on the solar zenith angle (SZA) (Eq. 1). However, the SZA does not appear explicitly in the objective function. The penalty on slant illumination conditions seems to be missing.

4. The top-level concept of the optimization scheme needs to be explained upfront (i.e. briefly in the abstract and in more detail in the introduction). Please clarify key elements such as a) that the scheme is to be applied off-line to determine a static scanning strategy, b) that near-real time information eg on cloud conditions is not taken into account, c) that the scheme is implemented by incrementally adding observation blocks, d) that the selection of the added blocks is performed by optimizing parameters X, Y, Z.

5. It is concluded that the IO based solution outperforms the "obvious human" solution. This statement needs to be either better supported addressing a the apparent weaknesses listed below, or revised. Weaknesses include: a) the "obvious human" solution is to some degree arbitrary, there might be better guesses; b) comparisons

are shown only for two cases with similar time of day, the situation can be different for other times; c) improvements in Amazonia and degradations at other places are reported (Fig 8), but it is not clear how global performances are determined and compared; d) differences in the total number of usable observations are reported (Section 4.1) but the basis of these numbers is unclear; How is the number of 'usable soundings' determined? Are there thresholds on SZA, AMF, SNR, albedo, .. ?

6. Section 4.2 reports a sensitivity analysis based on the assessment of regression coefficients. The conclusion of this analysis is unclear. Please clarify the conclusion in Section 4.2 and discuss the result in the overall context, in Section 5.

7. Figures 5 and 10 are not understood. Specify, also in the caption, which parameter is plotted on the ordinate, what the colour coding means, which distributions are represented by the 'violins'. Why are distributions plotted as double-sided graphs?

Technical Corrections

Section 3.6 The iterative determination of scanning blocks might be dependent on the starting point (the location of the first scanning block). Have various different starting positions been investigated?

Section 3.5.2 Is full and contiguous coverage of the continental Americas within +/- 50 deg lat within a day a hard boundary conditions for the optimization?

Page 4 line 18: the aerosol optical thickness of 0.3 is considered very large. Please justify. Aerosol optical depth depends on wavelength. What is the reference wavelength for the optical depth values provided?

Page 4 Eq 2: please provider units of parameters N0 and N1 (which should be same as the units of I)

Page 4 Eq 3: please clarify the meaning of sigma (introduced as the observational uncertainty). Clarify whether it is taken as the dominant contribution to the XCO2 vertical column uncertainty. Discuss the validity of this assumption.

[Figure]

Page 4 Eq 3: Specify units of sigma.

Page 5 Eq 4: eq 2 established a simple noise model. Eq 5 established an alternative more simplistic noise model. Why is the latter needed?

Page 5 line 7: unclear what is meant with "multiplicative inverse"

Page 5 Eq 5: 's' is used in an inconsistent way. It is introduced as an index to label the candidate block. It appears as a parameter in the argument AF, where it probably should not appear since x and y already capture the horizontal spatial dimensions. At the same time it represents an area in the spatial overlap operation; instead a dedicated variable (eg A_s) should be used to represent the area of the candidate block s.

Page 5 Eq 5: specify across which domain the median is evaluated. I guess it is the area of the candidate block 's'.

Page 5 Eq 5: The variables E and I should be introduced as 'areas of' the target land mass and of the selected scan blocks.

Page 5 Eq 5: the distance delta is not well defined. Please clarify from which point to which point is it to be evaluated.

Page 6 Section 3.5 discusses a finite number of possible locations of a scan block, which suggests that blocks can be located only at discrete positions. Please clarify whether this is correct. If so, introduce this constraint explicitly and specify the grid of candidate locations.

Section 3.5 Page 6 Section 3.5 line 9-10: The formulation ". . . a Greedy heuristic algorithm was employed to find a minimal covering set as a lower-bound estimate for set cardinality " is not understood. Please clarify what is meant with the term 'cardinality' in the present context?

Page 7 line 7-10 very long sentence, meaning is unclear. Please split and reformulate.

Page 7 line 7-10 Please clarify and elaborate how to the optimum at weights=1 is found.

Page 7 line 7-10 The variance of predicted errors is mentioned. On which parameter
and over which domain is this variance evaluated?

---

## Author Comment (AC1) · 26 Apr 2019

"This paper develops a scanning strategy for the upcoming GeoCarb mission from geostationary orbit, using an optimization algorithm to establish the greatest return in terms of soundings that exceed an unspecified minimum threshold in signal-to-noise ratio (SNR). The development of the approach appears logical and seems to give reasonable results. To me, though, it feels like a starting point for a more detailed treatment. It treats the land masses of the Americas as a "uniform space," treating all points as equally important. While it is crucial to obtain global coverage (over the viewing area of the satellite), it seems to me that not all locations are equivalent in terms of monitoring

greenhouse gas emissions. There are presumably hot spots of industrial activity that could benefit from closer scrutiny. The measurements do no extend to high enough latitude to capture emissions from the Alberta oil sands, but you will measure over the U.S. oil shale deposits in Colorado, Utah, and Wyoming, for example. I say this because one of the purported benefits of having the mission on a geostationary platform was that, according to the manuscript, "areas with high and uncertain anthropogenic emissions of $CO_2$, $CH_4$ and CO may be targeted with contiguous sampling," but this benefit is not exploited with the proposed scanning strategy. One would need to attribute increased weight to the hot spots to properly shade the coverage. The authors do mention the notion of adjusting the coverage to study events such as volcanoes and large wildfires, but that is a separate notion, a temporary campaign mode rather than a regular coverage strategy."

Our original goal for this study was to quantitatively obtain a scanning strategy that would cover the satellite viewing area once and result in the highest quality measurements of the Americas. The decision was made to scan the area between 50 degrees north and 50 degrees south in our study because it includes the areas of interest in the six science hypotheses stated in Moore et al (2018). To make this clearer to the reader, we've included the six hypotheses in the introduction.

We also agreed that demonstrating a scanning strategy with equal weighting for all land masses in the satellite viewing area does not illustrate to the reader the advantage of a geostationary platform. Therefore, we ran an additional experiment for a "city campaign" mode and added it to Section 5 of our manuscript. We would like to reiterate that this study is just a demonstration of one of many possible techniques and is not the proposed scanning strategy for the GeoCarb mission.

"Another benefit mentioned for the geostationary platform was that you could improve the signal by increasing the dwell time for a measurement. I do not know if there is some constraint that would require every scan to be identical. If one could select the measurement dwell time employed in each individual scan, for example, one could improve the results in regions where a large percentage of the soundings would otherwise fall below the SNR threshold and therefore be tagged as unusable. In this scenario, scan time would become an additional parameter to include in the optimization."

GeoCarb does not have the measurement dwell time as a parameter to be optimized since the long observation slit will, in general, cover areas of both low and high SNR in every scan. Additionally, maximizing observations from low SNR areas is not a primary goal of the mission, as seen in Moore et al (2018).

"The authors indicate that no results above the oceans are possible due to low signal. Is there no possibility of making use of ocean glint, as the OCO-2 mission does? This would obviously only work at certain times, when the conditions were such that sunlight reflected off the ocean at the same angle that the instrument was viewing the surface, but it would expand the coverage."

It is true that GeoCarb can theoretically make observations of ocean glint, but that is also not a primary objective of the mission.

"Judging from Figure 4, the gain in "usable soundings" relative to the baseline approach appears to be strongly related to a reduced number of measurements over the oceans. The expected improvement in errors over the Amazon is presumably related to the increased number of overlapping scans in that region."

In general, the increased number of overlapping scans is indeed a reason for increased number of soundings. However, we have added histogram plots of airmass and solar zenith angle, which are the stationary parameters on which our algorithm optimizes on, to show that the algorithm is selecting scanning blocks at peak airmass and solar zenith angle more than the baseline strategy.

"Note that no mention was made of what constituted a usable sounding. The last sentence relates the gain in soundings with SNR > 100, but it is not clear if that was the threshold employed for the determination of whether a particular measurement was usable. Based on the discussion on page 4, the calculated SNR was associated with the $O_2$ A-band measurements. That means you assumed the SNR for the measurements of the $CO_2$ bands was higher, or at the very least comparable?"

That was a typo and it has been corrected to say the SNR associated with the Weak $CO_2$ band.

"In the text (page 8, lines 9-10), the statement is made that Figure 6 showed the minimum error distribution medians and variances occur where both weights are equal to 1. For me, that does not seem like an obvious conclusion to draw from the figure. It certainly seems true for the upper left panel, and maybe for the median in the upper right panel, but unless I misinterpret what is being said, I do not see it for the other plots."

We changed our language to say that the weighting of the terms does not have a large impact on the predicted error. We point that the spread of error medians and variances is approximately 0.01 ppm and ultimately decided to leave the weighting at 1.

"Minor comments Page 2, line 2: the acronym "FoV" was defined but not used again, so there is no need to define the acronym"

Fixed in manuscript.

"Page 8, line 9: "Figure 6" should be "Fig. 6""

Since the figure is referenced at the beginning of the sentence, AMT guidelines tell us to spell out the entire word.

"Page 8, line 17: "Fig. 7 and 8" should probably be "Figs. 7 and 8""

Fixed in manuscript.

"Page 9, line 9: . . . mean Error. . . >Why is "Error" capitalized?"

Fixed in manuscript.

"Page 9, line 13: "Fig. 5 and 10" should probably be "Figs. 5 and 10""

Fixed in manuscript.

"Page 10, lines 4-5: "We also found that by optimizing for the global distribution of error, we obtained an improvement in regional errors as well, seen in Fig. 8" >This is not true for all regions, maybe an "overall improvement""

We agreed and fixed our language to state an "overall" improvement rather than the former.

"Page 10, line 13: the acronym "AOD" is not defined"

Now defined in manuscript.

"Figure 2: The caption claims that the plots relate to June (on the left) and December (on the right). The titles on the plots suggest they relate to September (on the left) and June (on the right)."

Fixed in manuscript.

"Figure 6: The variables w_dist and w_overlap are used to label the plots rather than w_d and w_o, the variables employed in the text."

Noted in caption.

"In Figures 8 and 9, it looks like there are no results over Cuba (greyed out?), even though that region appears to be within the scan range."

Cuba is not included in our scanning region as it is not a region of interest for the six hypotheses listed in Moore et al (2018).

―――――――――――――――

---

## Author Comment (AC2) · 27 Apr 2019

"The manuscript deals with the optimization of geographic coverage, which is a problem of interest for geostationary satellite remote sensing. The approach presented in the manuscript is new and deserves publication in AMT. However, the approach and its underlying assumptions need to be better explained and the discussion of the results needs to be made clearer. The manuscript needs to be revised addressing the issues identified below.

1. The proposed scheme aims at enhancing the yield and quality of a geostationary $CO_2$ observation system by optimizing the scanning strategy with a focus on the Signal

to Noise Ratio (SNR). Many other parameters that are expected to drive the yield and quality of such observations are not taken into account. Degraded CO2 product quality is expected not only in cases with low SNR, but also in many other conditions e.g. when viewing geometries are slant, when target air masses contain clouds or aerosol, and when clouds cover parts of the field of view thus increasing the risk of spatial straylight. The choice of focusing on SNR needs to be justified, and the approach regarding other potential drivers need to be explained and motivated."

We do include slant geometry into our calculation of SNR. We acknowledge that cloud contamination can cause bias in our measurements, but quantifying that effect is an open research topic and beyond the scope of this paper. The long viewing slit of GeoCarb means that at any given time there is a high probability that part of the slit will be obscured by clouds. Therefore, adaptive scanning to avoid clouds is beyond the scope of this first demonstration. Our algorithm seeks to maximize SNR by looking at the drivers that are more stationary processes such as the airmass and surface reflectance.

"2. The link between radiometric noise and the total CO2 uncertainty need to be discussed in more detail (beyond reference to O'Brian, 2016 and Eq. 3). The main contributors to the CO2 product uncertainty budget need to be discussed, and it needs to be explained why the optimization is driven by the random radiometric error."

The model we use is an empirical model of retrieval uncertainty as a function of SNR, not the actual physical models used in the L2 algorithm. We would like to point out that we are not minimizing total CO2 uncertainty, but rather uncertainty due to stationary processes such as the solar zenith angle, surface reflectance, and airmass. We have added additional explanation to Section 3.3 as to how we linked SNR to retrieved CO2 uncertainty.

"3. The objective function given (Eq. 5) minimized in the optimization scheme seems incomplete. The SNR depends on the radiance signal level (Eq. 2) hence also on the

solar zenith angle (SZA) (Eq. 1). However, the SZA does not appear explicitly in the objective function. The penalty on slant illumination conditions seems to be missing."

The penalty for slant illumination is contained in the airmass factor, m. As a matter of fact, we initially had SZA explicitly in our objective function. However, the viewing slit of the GeoCarb instrument is so long that the slant penalty accounted over the entire area of a scanning block can outweigh other penalties such as overlapping coverage. This would cause the algorithm to pick too many overlapping blocks and extend the scan past the usable daytime. Therefore, we found that the SZA accounted by the airmass factor was sufficient.

"4. The top-level concept of the optimization scheme needs to be explained upfront (i.e. briefly in the abstract and in more detail in the introduction). Please clarify key elements such as a) that the scheme is to be applied off-line to determine a static scanning strategy, b) that near-real time information e.g. on cloud conditions is not taken into account, c) that the scheme is implemented by incrementally adding observation blocks, d) that the selection of the added blocks is performed by optimizing parameters X, Y, Z."

a) The technique demonstrated in the paper was indeed applied offline and it has been noted in the manuscript. We would like to point out that this IO routine could be applied online with real-time information.

b) The technique demonstrated in the paper assumes cloud-free atmosphere. This has been specified in sections 2, 3, 3.3, and 4. We know that this is physically incorrect and we believe that, in the future, the IO routine can be modified to take in real-time cloud information if it were available for the entire scanning region. Although as previously mentioned, that area of research is beyond the scope of this paper. We would like to reiterate that this is just a demonstration of a technique and not the proposed scanning strategy for the GeoCarb mission.

c), d) Additional explanation of the main idea of IO has been added to the abstract and

introduction.

"5. It is concluded that the IO based solution outperforms the "obvious human" solution. This statement needs to be either better supported addressing a the apparent weaknesses listed below, or revised. Weaknesses include: a) the "obvious human"solution is to some degree arbitrary, there might be better guesses; b) comparisons are shown only for two cases with similar time of day, the situation can be different for other times; c) improvements in Amazonia and degradations at other places are reported (Fig 8), but it is not clear how global performances are determined and compared; d) differences in the total number of usable observations are reported (Section4.1) but the basis of these numbers is unclear; How is the number of 'usable soundings' determined? Are there thresholds on SZA, AMF, SNR, albedo, .. ?"

a) Prior to submitting this paper, the Moore et al (2018) had not been published. We possessed a tentative strategy and chose "obvious human" solution as a stand in for calling it the "proposed strategy" up until now. We have changed the language to say "proposed scanning strategy" rather than "obvious human" solution, referring to Moore et al (2018) as the source document for the GeoCarb mission description.

b) Part of the technique is that the algorithm chooses a timeframe for scanning by using a specified "starting airmass factor (AF) threshold" parameter. After specifying a starting AF threshold, the decision of when and where to start scanning is left to the algorithm. This is specified in Section 3.2.

c) Global performance in our context is meant to signify the aggregate predicted observational uncertainty (from Eq. (3) now Eq. (4)) over our satellite viewing area. We altered the language of the manuscript to clarify this.

d) We chose an SNR of 100 as our threshold as to what constitutes a usable sounding. In the empirical model of predicted $CO_2$ retrieval uncertainty as a function of SNR, a SNR of 100 translates to a 2 ppm XCO2 retrieval uncertainty, which is within the first proposed accuracy per sample of XCO2 mentioned in Polonsky et al (2014). We have

added extra explanation to the manuscript to clarify this.

"6. Section 4.2 reports a sensitivity analysis based on the assessment of regression coefficients. The conclusion of this analysis is unclear. Please clarify the conclusion in Section 4.2 and discuss the result in the overall context, in Section 5.7. Figures 5 and 10 are not understood. Specify, also in the caption, which parameter is plotted on the ordinate, what the colour coding means, which distributions are represented by the 'violins'. Why are distributions plotted as double-sided graphs?"

The x-axes of Figs. 5 and 10 are labeled as "starting threshold" in reference to the starting airmass factor threshold that the algorithm takes as a parameter. The colors are not in reference to any specific attribute of the distribution, rather it just makes it easier to distinguish between different distributions. We chose to represent our distributions as violins because we felt that it makes it easy for the reader to identify areas of high density within each distribution.

The sensitivity analysis was performed post-simulation runs as a check to see if the algorithm would exhibit unexpected behaviors when perturbed and concluded that it does not. These results are not tied to the main results of this paper. Therefore, the sensitivity analysis was moved to the appendix.

"Technical Corrections Section 3.6 The iterative determination of scanning blocks might be dependent on the starting point (the location of the first scanning block). Have various different starting positions been investigated?"

As mentioned in response to comment 5, the algorithm takes a specified starting air-mass factor threshold as an argument and then it decides when and where to start scanning. As a pseudo-sensitivity check early in conducting our experiments, we did try to force the algorithm to start at different areas, but it would return to scanning generally the same geographic locations as other algorithm-selected strategies within a short time after starting the scan.

"Section 3.5.2 Is full and contiguous coverage of the continental Americas within +/-50deg lat within a day a hard boundary conditions for the optimization?"

Yes. We have added additional explanation to the background section that explains that this geographic region includes the regions of interest for the six major science hypotheses stated in Moore et al (2018).

"Page 4 line 18: the aerosol optical thickness of 0.3 is considered very large. Please justify. Aerosol optical depth depends on wavelength. What is the reference wavelength for the optical depth values provided?"

We chose an aerosol optical thickness of 0.3 because it is considered a worst-case scenario for clear-sky retrievals and would give us conservative estimates of predicted observational uncertainty. This decision was based on the experience of the ACOS and OCO-2 team, referenced in the manuscript. We have added additional language clarifying that we are looking at the weak $CO_2$ (1.61 micron) band.

"Page 4 Eq 2: please provide units of parameters N0 and N1 (which should be sameas the units of I)"

Units have been added clarifying that I, N0 and N1 are in units of nW (cm^2 sr cm^(-1) )^(-1)

"Page 4 Eq 3: please clarify the meaning of sigma (introduced as the observational uncertainty). Clarify whether it is taken as the dominant contribution to the XCO2vertical column uncertainty. Discuss the validity of this assumption."

Additional explanation of sigma was added to Section 3.3, which explains that sigma is derived from the posterior covariance given by the L2 algorithm.

"Page 4 Eq 3: Specify units of sigma."

The manuscript has been fixed to say that sigma is in units of ppm.

"Page 5 Eq 4: eq 2 established a simple noise model. Eq 5 established an alternative

more simplistic noise model. Why is the latter needed?"

The more simplified model in Eq. 5 is an intermediate step to explaining the formulations of the objective function. It has been moved to be an inline equation rather than a numbered equation block.

"Page 5 line 7: unclear what is meant with "multiplicative inverse""

Additional language was added to clarify that we mean, one divided by the radiance.

"Page 5 Eq 5: 's' is used in an inconsistent way. It is introduced as an index to label the candidate block. It appears as a parameter in the argument AF, where it probably should not appear since x and y already capture the horizontal spatial dimensions. At the same time it represents an area in the spatial overlap operation; instead a dedicated variable (eg A_s) should be used to represent the area of the candidate block s."

We realized that having set operations in the equation could be ambiguous to the reader. The terms in the objective function were reformulated to exclude set operations.

"Page 5 Eq 5: specify across which domain the median is evaluated. I guess it is the area of the candidate block 's'."

Additional language was added to clarify that we meant the area of the candidate block, s.

"Page 5 Eq 5: The variables E and I should be introduced as 'areas of' the target landmass and of the selected scan blocks."

The terms in the objective function were reformulated to be more clear to the reader.

"Page 5 Eq 5: the distance delta is not well defined. Please clarify from which point to which point is it to be evaluated."

Additional language was added to clarify that delta is the shortest linear distance from the boundary of the last selected scan block to the candidate scan block. A diagram

has also been added to clarify the terms of distance, overlap, and coverage.

"Page 6 Section 3.5 discusses a finite number of possible locations of a scan block, which suggests that blocks can be located only at discrete positions. Please clarify whether this is correct. If so, introduce this constraint explicitly and specify the grid of candidate locations."

Additional explanation was added to Section 3.5, referencing the explanation of the formulation of scan blocks in Section 3.1.

"Section 3.5 Page 6 Section 3.5 line 9-10: The formulation "...a Greedy heuristic algorithm was employed to find a minimal covering set as a lower-bound estimate for set cardinality " is not understood. Please clarify what is meant with the term 'cardinality' in the present context?"

The term 'cardinality' has been changed to say "set size".

"Page 7 line 7-10 very long sentence, meaning is unclear. Please split and reformulate. Page 7 line 7-10 The variance of predicted errors is mentioned. On which parameter and over which domain is this variance evaluated?"

We believe that the commenter is referencing to Page 8 lines 7-10. The entire Section 3.7 (now Sect. 3.6) has been reformulated for clarity.

"Page 7 line 7-10 Please clarify and elaborate how to the optimum at weights=1 is found."

We changed our language to say that the weighting of the terms have negligible effects on the predicted error. We point that the spread of error medians and variances is approximately 0.01 ppm and ultimately decided that the weights shall remain equal to 1.

---

## Author Response (AR2)

Comments to the Author:

Thank you for you attention to the referee comments. I have read your revisions and consider the paper ready for publication following correction to the following:

page 5: "... errors in retrieved gases /are/ arise from ..."

delete /are/ ...

Response to the editor:

Corrections have been made in accordance with your suggestions.

Page 5, line 2: (deleted) "Errors in retrieved gases (are) …"

Page 5, line 2: (added)   "Errors in retrieved gases (arise from) …"